# Association of Asymmetric Dimethylarginine and Diastolic Dysfunction in Patients with Hypertrophic Cardiomyopathy

**DOI:** 10.3390/biom9070277

**Published:** 2019-07-13

**Authors:** Kathrin Cordts, Doreen Seelig, Natalie Lund, Lucie Carrier, Rainer H. Böger, Maxim Avanesov, Enver Tahir, Edzard Schwedhelm, Monica Patten

**Affiliations:** 1Institute of Clinical Pharmacology and Toxicology, University Medical Center Hamburg-Eppendorf, 20246 Hamburg, Germany; 2Clinic of General and Interventional Cardiology, University Heart Center, 20246 Hamburg, Germany; 3Institute of Experimental Pharmacology and Toxicology, University Medical Center Hamburg-Eppendorf, 20246 Hamburg, Germany; 4DZHK (German Centre of Cardiovascular Research), partner site Hamburg/Kiel/Lübeck, 20246 Hamburg, Germany; 5Clinic of Diagnostic and Interventional Radiology and Nuclear Medicine, University Medical Center Hamburg-Eppendorf, 20246 Hamburg, Germany

**Keywords:** arginine derivatives, hypertrophic cardiomyopathy, biomarker, diastolic dysfunction

## Abstract

Despite genetic heterogeneity, early manifestation of diastolic dysfunction (DD) is common in hypertrophic cardiomyopathy (HCM). Nitric oxide (NO) may contribute to myocardial relaxation. NO synthases (NOS) use l-arginine (Arg) as a substrate, as asymmetric dimethylarginine (ADMA) is a direct endogenous inhibitor of NOS. This study aimed to analyze the association of Arg and its derivates, i.e., l-homoarginine (hArg), ADMA and symmetric dimethylarginine (SDMA), with DD in HCM patients. In 215 HCM patients (mean age 54 ± 15 years, 58% male) transmitral and mitral annulus velocities were echocardiographically analyzed. Plasma concentrations of Arg derivatives were measured by liquid chromatography tandem-mass spectrometry. In 143 (70%) patients suffering from DD, ADMA showed the strongest association with DD (0.66 ± 0.16, 0.72 ± 0.24, and 0.76 ± 0.26 µmol/L, *p* < 0.01 for trend). In linear regression analyses, positive association per standard deviation increase of ADMA was found with E-wave (beta coefficient (95% confidence interval): 4.72 (0.43–9.01); *p* < 0.05) and mean E/E’ (1.76 (0.73–2.79) *p* < 0.001). Associations were adjusted for age, sex, body mass index (BMI), diabetes mellitus, coronary artery disease, and arterial hypertension. Elevated ADMA is associated with the severity of DD in HCM. Higher ADMA level might lead to decreased NO production and thus an impaired myocardial relaxation pattern.

## 1. Introduction

Hypertrophic cardiomyopathy (HCM) is an inherited disease, characterized by an asymmetrical hypertrophy of the left ventricle (LV) [1]. In adults HCM occurs with a prevalence of 0.02–0.23%, most commonly caused by autosomal dominant inherited mutations in genes that encode cardiac sarcomere proteins. Despite genetic heterogeneity, early manifestation of diastolic dysfunction (DD) is common. Nitric oxide (NO) is an important anti-atherosclerotic mediator of vascular function and microvascular dysfunction is an early companion of DD in HCM patients [2,3]. In the heart, NO plays a role for ion channel activity, thereby regulating cardiac contraction as well as dilation [4]. In diastole, NO induces earlier, faster relaxation, and reduced left ventricular pressure [5]. NO is synthesized from l-arginine (Arg), catalyzed by NO synthases, comprising endothelial, neuronal, and inducible isoforms [6,7]. NO synthesis is directly inhibited by the Arg analogue asymmetric dimethylarginine (ADMA) and indirectly by symmetric dimethylarginine (SDMA), which inhibits the cellular uptake of arginine [8,9]. In contrast, l-homoarginine (hArg) increases bioavailability of NO by serving as a substrate for NO synthases and by inhibiting degradation of Arg by arginases [10]. Pilz et al. found an association of decreased hArg and increased ADMA and SDMA with DD in a large cohort of primary care patients [11]. However, the biological role of homoarginine is not clear to date. An inhibitory effect of homoarginine on arginase activity could be demonstrated in association with lower cardiovascular mortality, but not in physiological concentrations [12]. We hypothesized that impaired NO production might augment impaired cardiac relaxation in HCM patients. Therefore, we investigated possible associations of Arg and its derivatives, i.e., ADMA, SDMA and hArg, with the echocardiographically assessed presence and severity of DD in HCM patients as the primary endpoint. The secondary end-point represents a possible association of Arg and its derivatives with echocardiographically assessed left ventricular hypertrophy, presence of atrial fibrillation and fibrosis determined in cardiac magnetic resonance (CMR) imaging.

## 2. Methods

### 2.1. Study Population

A total of 215 HCM patients (age 18–99 years), enrolled from routine visits in our outpatient clinic at the University Heart Center Hamburg-Eppendorf between May 2011 and September 2016, were included in this retrospective cross-sectional study. Medical histories were recorded, a 12 lead-ECG was performed, and patients were examined physically and by echocardiography. According to current guidelines, HCM was defined as a maximum wall thickness of ≥15 mm in the absence of abnormal loading conditions or another cardiac or systemic disease that could produce the magnitude of hypertrophy evident. In 56 patients, HCM was genetically confirmed. Exclusion criteria were pregnancy and an estimated glomerular filtration rate (eGFR) <30 mL/min/1.73 m^2^. The study protocol was in line with the principles outlined in the Declaration of Helsinki and approved by the local ethics committee (Ärztekammer Hamburg, PV4056). All patients gave their written informed consent.

### 2.2. Clinical Evaluation

During routine visits a 12-lead surface electrocardiogram (ECG; CS-200, Schiller Inc., Baar, Switzerland) and a standard echocardiogram (iE33, Philips, Amsterdam, The Netherlands) were assessed. Serum and plasma samples were drawn for routine laboratory parameters (serum creatinine and creatine kinase). eGFR was calculated using the modification of diet in renal disease (MDRD)-formula. Medical history of coronary artery disease (CAD), diabetes mellitus and arterial hypertension was confirmed by self-report or the use of corresponding drugs. Atrial fibrillation (AF) was assessed by positive history within the last five years from examination. Patients were symptomatically classified according to the New York Heart Association (NYHA)-classification.

### 2.3. Quantification of l-Arginine Derivatives

Arg, ADMA, SDMA and hArg were measured in plasma samples by liquid chromatography-tandem mass spectrometry (LC-MS/MS) [13,14]. In brief, 25 µL plasma were spiked with stable isotope labelled internal standards solved in 100 µL methanol, proteins were precipitated, and samples were filtrated through a 0.22 μm hydrophilic membrane (Multiscreen HTS, Millipore, Molsheim, France). Methanol was evaporated at 80 °C and analytes were derivatized to their butyl ester derivatives by incubating samples with butanolic 1 mol/L HCl for 30 min at 65 °C. After evaporation at 75 °C, analytes were reconstituted in sample buffer (methanol–water (50:50, *v*/*v*) containing 0.1% formic acid, pH 4.95 for Arg, ADMA and SDMA; methanol–water (25:75, *v*/*v*) containing 0.1% formic acid, pH 4.95 for hArg). LC-MS/MS analyses were performed on a Varian 1200 MS equipped with two Varian ProStar model 210 HPLC pumps (Agilent Technologies, Santa Clara, USA).

### 2.4. Echocardiography

To assess parameters of diastolic function and hypertrophy two-dimensional (2D) transthoracic echocardiography was performed on a Philips iE33 system (Philips Healthcare, Best, Netherlands) and data were analyzed by Syngo Dynamics (Siemens Healthcare, Erlangen, Germany). 2D apical images were used to obtain the left ventricular ejection fraction (LVEF) applying the Simpson method. Patients with LVEF ≤55% were classified as patients with a reduced LVEF [15]. Left atrial diameter, septal (SW) and lateral wall (LW) thickness were obtained from measurements in parasternal short axis. Pulsewave Doppler of transmitral flow in the apical 4-chamber view was used to obtain peak early (E-wave) and late (A-wave) transmitral filling velocities. Tissue Doppler imaging (TDI) was used in the apical 4-chamber view to measure early septal and lateral mitral annulus velocities (E’) and septal and lateral isovolumic relaxation time (IVRT). DD was classified according to Nagueh et al. 2009 [16]: Normal diastolic function (septal E‘ ≥ 8 cm/s, lateral E‘ ≥ 10 cm/s, left atrial volume <34 mL/m^2^), mild DD (grade 1; septal E‘ < 8 cm/s, lateral E‘ < 10 cm/s, left atrial volume ≥ 34 mL/m^2^, E/A < 0.8, E/E‘ ≤ 8), moderate DD (grade 2; septal E‘ < 8 cm/s, lateral E‘ < 10 cm/s, left atrial volume ≥ 34 mL/m^2^, E/A 0.8–1.5, E/E‘ 9–12), and severe DD (grade 3; septal E‘ < 8 cm/s, lateral E‘ < 10 cm/s, left atrial volume ≥ 34 mL/m^2^, E/A ≥ 2, E/E‘ ≥ 13). DD classification was available for 204 patients (95%). Resting and provoked left ventricular outflow tract (LVOT) flow gradients were assessed and patients classified as non-obstructive hypertrophic cardiomyopathy ((HNOCM; resting LVOT gradient <30 and provoked LVOT gradient <30 mmHg), latent obstructive hypertrophic cardiomyopathy (HLOCM; resting LVOT gradient <30 and provoked LVOT gradient >50 mmHg), and obstructive hypertrophic cardiomyopathy (HOCM; resting LVOT gradient >30 mmHg).

### 2.5. Cardiac Magnetic Resonance Imaging

CMR was performed using a 1.5 T Achieva scanner equipped with a 5-channel cardiac phased array receiver coil (Philips Healthcare, Best, The Netherlands) in 129 HCM patients. The CMR protocol included late gadolinium enhancement (LGE) imaging using a phase-sensitive inversion recovery (PSIR) sequence. Gadoter acid (Dotarem, Guerbet, Sulzbach, Germany) was administered as bolus injection of 0.2 mmol/kg at a rate of 2.5 mL/s and after ten minutes end-diastolic LGE images were acquired: AVS 1.59 × 1.71 × 8 mm^3^, RVS 0.97 × 0.98 × 8 mm^3^, gap 2 mm, 9–10 slices, echo time = 2.40 ms, time to repetition = 5.50 ms, flip angle = 15°. The optimal inversion delay was obtained from a Look-Locker experiment. LGE images were acquired in short-axis orientation covering the entire heart and in two-, three- and four-chamber views. 

Measurements were performed by two independent radiologists who were blinded to clinical data. Myocardial contraction fraction was calculated according to Arenja et al. 2017 [17]. For statistical analysis the amount of LGE and the myocardial contraction fraction (MCF) were determined.

### 2.6. Statistical Analysis

Normally distributed continuous data are given as mean ± standard deviation (SD), and otherwise as median (25–75th percentile). Categorical data are given as n-number of participants (percentage). Data not normally distributed were log-transformed before analysis if necessary. Two groups were compared using student’s *t*-test, χ^2^ test or logistic regression analyses (odds ratio (OR) and 95% confidence interval (CI)). More than two groups were compared using one-way analysis of variance (ANOVA) with post test for linear trend, or analysis of covariance (ANCOVA), adjusted for eGFR. Associations of continuous data were analyzed by Spearman, Pearson and partial correlation analyses (correlation coefficient ρ), or linear regression analyses (beta coefficient and 95% CI). Logistic and linear regression analyses were calculated for four models: model 1 was unadjusted, model 2 was adjusted for age and sex, model 3 was additionally adjusted for body mass index (BMI), diabetes mellitus, CAD and arterial hypertension, and model 4 was additionally adjusted for eGFR. Partial correlation analyses were adjusted for age, sex, BMI, diabetes mellitus, CAD, arterial hypertension and eGFR. A *p*-value < 0.05 was considered to be statistically significant. Statistical analyses were performed using IBM SPSS Statistics (version 22, IBM Corp., Armonk, USA) and GraphPad Prism (version 5 for Windows, La Jolla, USA).

## 3. Results

A total of 215 HCM patients were included in this observational study. The baseline characteristics, echocardiographic parameters and cardiac magnetic resonance imaging parameters of all HCM patients, and stratified by diastolic function are depicted in Table 1. 

### 3.1. Arginine Derivatives and Diastolic Dysfunction

Stratification of HCM patients according to the presence of DD revealed that those suffering from DD had higher ADMA (0.74 ± 0.25 vs. 0.66 ± 0.16 µmol/L; *p* = 0.003), higher SDMA (0.59 ± 0.24 vs. 0.52 ± 0.14 µmol/L; *p* = 0.008) and lower hArg (1.54 ± 0.68 vs. 1.76 ± 0.64 µmol/L; *p* = 0.031), whereas Arg was not different compared to HCM patients with a normal diastolic function (52 ± 38 vs. 50 ± 21 µmol/L; *p* = 0.575). Stratified by the grades of diastolic dysfunction, ADMA concentration increased with increasing severity of DD (Table 2). eGFR-adjusted ANCOVA revealed a differentiation of patients with a normal diastolic function and patients with a moderate to severe DD (*p* = 0.026). No differences were found for SDMA and hArg. In logistic regression analyses, ADMA showed an association when plasma concentrations of patients with a moderate to severe DD were compared to patients with a normal diastolic function (models 1, 3, and 4; Table 3). In the subgroup of the genetically confirmed HCM-patients ADMA concentration also increased with increasing severity of DD (Appendix A). In unadjusted regression analyses ADMA concentration showed an association when comparing genetically confirmed HCM patients with a moderate to severe DD, with patients having a normal diastolic function (Appendix A).

In Spearman correlation analyses of arginine derivatives with echocardiographic parameters of diastolic function, ADMA was positively correlated with the E-wave (ρ = 0.18; *p* = 0.008; Appendix A) and mean E/E’ (ρ = 0.23; *p* < 0.001) and correlated inversely with the mean E’ (ρ = −0.19; *p* = 0.007). SDMA was positively associated with the E-wave (ρ = 0.15; *p* = 0.032) and mean E/E’ (ρ = 0.17; *p* = 0.019). hArg was positively associated with mean E‘ (ρ = 0.17; *p* = 0.019), and inversely with mean IVRT (ρ = −0.19; *p* = 0.008). No associations were found in Pearson correlation and partial correlation analyses (Appendix A). In linear regression analyses, positive associations that remained significant after adjustment for possible confounders were found for ADMA and E-wave (model 1: beta coefficient (95% confidence interval); 4.90 (0.79–9.01); model 2: 4.58 (0.35–8.80); model 3: 4.72 (0.43–9.01); model 4: 4.67 (0.37–8.98); *p* < 0.05 for all; Appendix A) and mean E/E´ (model 1: 2.24 (1.17–3.31); model 2: 1.80 (0.77–2.83); model 3: 1.76 (0.73–2.79); model 4: 1.77 (0.74–2.80); *p* < 0.001 for all). No associations were found for hArg and SDMA. Correlation and linear regression analyses were not reproducible in the small subgroup of genetically confirmed HCM patients.

### 3.2. Arginine Derivatives and Atrial Fibrillation

HCM patients who experienced AF within the last five years had higher plasma ADMA (0.77 ± 0.21 vs. 0.69 ± 0.23 µmol/L; *p* = 0.011; mean ± SD; student’s *t*-test; *n* = 67 vs. 148), higher SDMA (0.66 ± 0.23 vs. 0.53 ± 0.20 µmol/L; *p* < 0.001), and lower hArg (1.42 ± 0.54 vs. 1.69 ± 0.70 µmol/L; *p* = 0.002), whereas Arg did not differ (48 ± 23 vs. 54 ± 38 µmol/L; *p* = 0.224). In logistic regression analyses, no association of AF was found for ADMA, but for SDMA and hArg and AF in unadjusted and adjusted models (Table 4). After further adjustment for eGFR, the association of SDMA and AF missed statistical significance. No associations of arginine derivatives were found in adjusted logistic regression analyses in the subgroup of genetically confirmed HCM patients (Appendix A).

### 3.3. Arginine Derivatives, Cardiac Hypertrophy, and Fibrosis

In Spearman correlation analyses, hArg was inversely associated with SW thickness (ρ = −0.23; *p* < 0.001; Appendix A), whereas no association was found for LW thickness (ρ = 0.003; *p* = 0.966). No associations were seen for ADMA (ρ= −0.12; *p* = 0.073 for septal wall thickness and ρ = 0.05; *p* = 0.517 for lateral wall thickness) and SDMA (ρ = −0.13; *p* = 0.061 for SW thickness and ρ = 0.03; *p* = 0.719 for LW thickness). No associations were found in Pearson correlation and partial correlation analyses (Appendix A). Plasma Arg derivatives concentrations did not differ between HCM patients with or without fibrosis (ADMA: 0.72 ± 0.24 vs. 0.74 ± 0.16 µmol/L; *p* = 0.700; SDMA: 0.54 ± 0.18 vs. 0.61 ± 0.17 µmol/L; *p* = 0.178; hArg: 1.55 ± 0.63 vs. 1.93 ± 0.69 µmol/L; *p* = 0.102; Arg: 52 ± 41 vs. 61 ± 25 µmol/L; *p* = 0.265; *n* = 11 vs. 114; mean ± SD; Student’s *t*-test). Myocardial contraction fraction (MCF) serves both as a marker and index for fractional myocardial shortening [17]). MCF is calculated by dividing the SV stroke volume by the LV myocardial volume, multiplied by 100%. This calculation is easily derived from imaging and is therefore a good marker for the diagnosis of patients with left ventricular hypertrophy [17]. In Spearman correlation analyses Arg derivatives were not associated with MCF (ρ = −0.03, *p* = 0.750 for ADMA; ρ = 0.05, *p* = 0.556 for SDMA; ρ = −0.02, *p* = 0.818 for hArg, and ρ = 0.06, *p* = 0.488 for Arg). 

## 4. Discussion

### 4.1. Arginine Derivatives and Diastolic Dysfunction

Among all Arg derivatives investigated, we found the strongest associations for ADMA. Stratified by the grades of diastolic dysfunction, ADMA concentration was elevated with increasing severity of DD (*p* < 0.01; ANOVA with post-test for linear trend). In a large cohort of primary care patients, ADMA and SDMA concentrations were higher and hArg concentrations were lower in DD patients compared to patients with normal diastolic function [11]. In this study lower hArg was independently associated with the presence of DD and plasma ADMA was increasing with the severity of DD. In linear regression analyses we found positive association per standard deviation (SD) increase of ADMA, with E-wave and mean E/E´ (Appendix A). Accordingly, negative associations were found for mean E´. Associations for E-wave and mean E/E´ remained significant after adjustment for age, sex, BMI, diabetes mellitus, coronary artery disease and arterial hypertension. Myocardial relaxation is at least partly dependent on NO [5]. In experimental studies NO enhanced cardiac relaxation without affecting systolic parameters [18,19]. In HCM patients an increase of the vascular endothelial growth factor (VEGF) is related to impaired ejection fraction and fractional shortening of the LV [20]. Moreover, local administration of NO improves LV relaxation and diastolic distensibility [21]. In heart failure, patient accumulation of ADMA and SDMA was associated with echocardiographic parameters of DD, indicating an association between DD and an impaired NO metabolism [22]. In these patients, ADMA showed the strongest association with disease progression and adverse long-term outcomes. In our present study we also found the strongest associations of ADMA with echocardiographic parameters of DD and the severity of DD in HCM patients. Further studies are needed to evaluate the mechanistic role of ADMA for the pathogenesis of DD in HCM patients and a possible prognostic value for the long-term outcome of these patients.

### 4.2. Arginine Derivatives and Atrial Fibrillation

DD is a risk factor for AF, which is the most common form of arrhythmia in HCM patients [1]. In our HCM cohort, decreased hArg was independently associated with the presence of AF within the last five years prior to inclusion in this study. However, in the community-based cohort of the Framingham Heart Study, ADMA and SDMA were not independently associated with the incidence of AF [23]. In the general population of the Gutenberg Health Study higher ADMA was associated with prevalent AF, whereas no association was seen for SDMA or hArg [24,25]. In CAD patients, lower hArg was associated with a higher prevalence of AF and an increased risk for incident fatal and non-fatal stroke or MI and all-cause mortality [26]. Elevated ADMA and SDMA were associated with increased mortality in AF patients [27]. In AF, formation of thromboembolism is increased, leading to a higher risk of sudden cardiac death, heart failure and stroke [28]. Further studies are needed to evaluate possible mechanistic or prognostic values of modulators of the NO bioavailability (i.e., ADMA, SDMA and hArg) in AF patients.

### 4.3. Arginine Derivatives and Cardiac Hypertrophy

In HCM induced by sarcomeric gene mutation, an asymmetrical hypertrophy mainly of the left ventricular SW is common [1]. Some studies have found associations of Arg derivatives with the presence of LV hypertrophy in cardiac patients [11,29]. We therefore investigated associations of Arg derivatives with echocardiographically assessed LV hypertrophy (i.e., SW and LW) and fibrosis (assessed in CMR) in our HCM patient cohort. Only hArg was inversely associated with SW thickness. In vivo hArg is produced by the l-argininine:glycine amidinotransferase (AGAT), which is furthermore the first enzyme for endogenous creatine synthesis and is therefore involved in energy metabolism [30]. In heart failure, patient myocardial AGAT expression is upregulated, indicating a local creatine synthesis in the energy-starved heart [31]. However, myocardial AGAT expression is, compared to the kidney and liver, relatively low and might not affect plasma hArg concentration to a great extent. In contrast, we found lower hArg with increasing SW, but no association with fibrosis in our HCM patients. In recent years, studies have been conducted that linked increased ADMA concentration to the development of hypertrophy [32,33]. Nevertheless, the exact genesis could not be clarified, but a change of the nitric oxide metabolism is suspected [32,34,35]. Further experimental studies are needed to evaluate if there is a causal link between hArg or ADMA and hypertrophy in HCM patients.

### 4.4. Study Limitations

A relatively small sample size of HCM patients was investigated; only 56 out of 215 were genetically confirmed HCM patients, and primary and secondary end-points were restricted to LV function and structure (excluding vascular and inflammatory biomarkers). Furthermore, we want to point out that this cross-sectional single center study is an observational study, and so therefore we cannot make any conclusions about the causality of our findings.

## 5. Conclusions

Among all l-arginine derivatives investigated, we found the strongest associations for ADMA and the severity of DD and related echocardiographic parameters of cardiac relaxation. As a direct inhibitor of NOS, higher ADMA plasma concentrations might lead to a decreased NO production and thus to an impaired myocardial relaxation pattern in HCM patients. Lower hArg was associated with AF in our cohort, as well as inversely associated with SW thickness. Further studies are needed to confirm the results and clarify if there is a mechanistic link between the investigated Arg derivatives and the pathogenesis of DD, AF and cardiac hypertrophy in HCM patients.

## Figures and Tables

**Table 1 biomolecules-09-00277-t001:** Baseline characteristics.

	HCM (*n* = 215)	HCM no DD (*n* = 61)	HCM DD (*n* = 143)	*p*-Value
Age (years)	54 ± 15	44 ± 15	58 ± 13	<0.001
Sex (males, *n*)	125 (58.1%)	43 (70.5%)	75 (52.4%)	0.017
BMI (kg/m^2^)	27 ± 5	25 ± 4	28 ± 6	<0.001
AF (medical history) (*n*)	67 (31.2%)	11 (18.0%)	49 (34.3%)	0.020
NYHA (*n*)				<0.001
I-II	145 (67.4%)	56 (91.8%)	81 (56.6%)	
III-IV	70 (32.6%)	5 (8.2%)	62 (43.4%)	
Creatinine (mg/dL)	0.99 ± 0.27	0.93 ± 0.25	1.01 ± 0.28	0.034
eGFR (mL/min)	81 ± 26	94 ± 27	76 ± 23	<0.001
QTc (ms)	438 ± 33	422 ± 27	444 ± 34	<0.001
**Concomitant Diseases**				
Diabetes mellitus	25 (12%)	4 (6.6%)	19 (13.3%)	0.164
Coronary artery disease	36 (17%)	4 (6.6%)	28 (19.6%)	0.019
Arterial hypertension	96 (45%)	18 (29.5%)	74 (51.7%)	0.003
Echocardiographic parameters				
SW thickness (mm)	22 ± 5	21 ± 6	22 ± 5	0.050
LW thickness (mm)	14 ± 4	13 ± 4	15 ± 3	0.002
Obstruction (*n*)				0.005
HNOCM	71 (47.7%)	24 (68.6%)	45 (40.9%)	
HLOCM	19 (12.8%)	5 (14.3%)	13 (11.8%)	
HOCM	59 (39.6%)	6 (17.1%)	52 (47.3%)	
**Diastolic Function**				
No DD	61 (29.9%)	61 (29.9%)	0 (0%)	
Mild DD	50 (24.5%)	0 (0%)	50 (24.5%)	
Moderate DD	86 (42.2%)	0 (0%)	86 (42.2%)	
Severe DD	7 (3.4%)	0 (0%)	7 (3.4%)	
Reduced LVEF (*n*)	20 (9.4%)	2 (3.3%)	16 (11.2%)	0.068
E-wave (m/s)	93 ± 29	87 ± 22	95 ± 32	0.036
A-wave (m/s)	79 ± 54	60 ± 20	88 ± 61	<0.001
E/A	1.37 ± 0.68	1.55 ± 0.57	1.28 ± 0.71	0.007
Mean E‘(cm/s)	6.8 ± 2.2	8.9 ± 1.9	5.9 ± 1.7	<0.001
Mean E/E‘	16.1 ± 7.7	10.6 ± 2.8	18.7 ± 7.9	<0.001
Mean IVRT (ms)	123 ± 35	100 ± 20	134 ± 35	<0.001
Left atrial diameter (mm)	46 ± 10	41 ± 7	48 ± 10	<0.001
**Cardiac Magnetic Resonance Imaging Parameters**				
MCF (%)	60 ± 27	65 ± 23	60 ± 28	0.312
LGE positive (*n*)	114 (91.2%)	36 (87.8%)	71 (92.2%)	0.433

Data are given as mean ± standard deviation (SD), median (25–75th percentile), or n-number (percentage). Student’s t test or chi-square test were applied to compare HCM patients with normal diastolic function vs. diastolic dysfunction. AF, atrial fibrillation; BMI, body mass index; DD, diastolic dysfunction; eGFR, estimated glomerular filtration rate; HCM, hypertrophic cardiomyopathy; HLOCM, latent obstructive HCM; HNOCM, non-obstructive HCM; HOCM, obstructive HCM; IVRT, isovolumetric relaxation time; LGE, late gadolinium enhancement; LVEF, left ventricular ejection fraction; LW, lateral wall; MCF, myocardial contraction fraction; NYHA, New York Heart Association; QTc, corrected QT interval; SW, septal wall.

**Table 2 biomolecules-09-00277-t002:** Arginine derivatives in HCM-patients stratified by diastolic function.

	HCM No DD	HCM Mild DD	HCM Moderate/Severe DD	*p*-Value for Linear Trend
**ADMA (µmol/L)**	0.66 ± 0.16	0.72 ± 0.24	0.76 ± 0.26	0.006
**SDMA (µmol/L)**	0.52 ± 0.14	0.61 ± 0.29	0.58 ± 0.20	0.074
**hArg (µmol/L)**	1.76 ± 0.64	1.52 ± 0.63	1.56 ± 0.71	0.069

Data are given as mean ± standard deviation (SD); one-way analysis of variance (ANOVA) with post-test for linear trend. ADMA indicates asymmetric dimethylarginine; Arg, arginine; DD, diastolic dysfunction; hArg, homoarginine; HCM, hypertrophic cardiomyopathy; SD, standard deviation; SDMA, symmetric dimethylarginine.

**Table 3 biomolecules-09-00277-t003:** Logistic regression analyses for the arginine derivatives and diastolic dysfunction.

	ADMA		SDMA		hArg	
	OR (95% CI)	*p*-Value	OR (95% CI)	*p*-Value	OR (95% CI)	*p*-Value
HCM patients (moderate to severe DD vs. normal diastolic function)						
**Model 1**	1.79 (1.14–2.79)	0.011	1.54 (0.99–2.40)	0.056	0.76 (0.55–1.05)	0.100
**Model 2**	1.53 (0.93–2.51)	0.096	1.12 (0.70–1.79)	0.624	0.86 (0.61–1.22)	0.408
**Model 3**	1.76 (1.07–2.89)	0.026	1.31 (0.79–2.16)	0.292	0.72 (0.49–1.06)	0.097
**Model 4**	1.87 (1.12–3.12)	0.016	1.14 (0.68–1.94)	0.615	0.73 (0.49–1.08)	0.116

Logistic regression analyses with odds ratios (95% CI) per SD increase. Model 1 is unadjusted, model 2 is adjusted for age and sex, model 3 is additionally adjusted for body mass index, diabetes mellitus, coronary artery disease, and arterial hypertension, model 4 is additionally adjusted for estimated glomerular filtration rate. ADMA, asymmetric dimethylarginine; CI, confidence interval; hArg, homoarginine; OR, odds ratio; SD, standard deviation; SDMA, symmetric dimethylarginine.

**Table 4 biomolecules-09-00277-t004:** Logistic regression analyses of arginine derivatives and atrial fibrillation.

	ADMA		SDMA		hArg	
	OR (95% CI)	*p-*Value	OR (95% CI)	*p-*Value	OR (95% CI)	*p*-Value
**Model 1**	1.34 (0.99–1.82)	0.060	1.91 (1.35–2.69)	<0.001	0.62 (0.44–0.87)	0.006
**Model 2**	1.08 (0.79–1.49)	0.620	1.50 (1.06–2.14)	0.024	0.69 (0.48–0.98)	0.039
**Model 3**	1.05 (0.75–1.46)	0.781	1.51 (1.05–2.19)	0.027	0.66 (0.46–0.95)	0.027
**Model 4**	1.05 (0.76–1.47)	0.751	1.39 (0.95–2.03)	0.094	0.68 (0.47–0.98)	0.038

Logistic regression analyses with odds ratios (95% CI) per SD increase. Model 1 is unadjusted, model 2 is adjusted for age and sex, model 3 is additionally adjusted for body mass index, diabetes mellitus, coronary artery disease, and arterial hypertension, model 4 is additionally adjusted for estimated glomerular filtration rate. ADMA, asymmetric dimethylarginine; Arg, arginine; CI, confidence interval; hArg, homoarginine; OR, odds ratio; SDMA, symmetric dimethylarginine.

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
