# Peer review of "Association of Asymmetric Dimethylarginine and Diastolic Dysfunction in Patients with Hypertrophic Cardiomyopathy"

_biomolecules, 2019, doi:10.3390/biom9070277_

Round 1

Reviewer 1 Report

There are several issues that require attention:

Introduction: should spell out that the ADMA and SDMA have diverse effects on NO synthesis.

Introduction: the biological role of homoarginine, e.g. arginase inhibition, is far from established (e.g. Tommasi S et al, Sci Rep 2018;8(1): 3697). This should be discussed.

Methods, CMR: this paragraph should spell out which specific parameters were measured for the purpose of the analysis.

Methods: the authors should clearly describe the study hypothesis and list the primary and secondary end-points of cardiac structure and function.   

A key issue that limits data interpretation is the lack of adjustment for eGFR. ADMA and, particularly, SDMA undergo renal elimination. I note that eGFR is higher in no DD than DD patients (Table 1). All regression and ANOVA analyses should also be adjusted for eGFR. 

Present all correlation analyses in Table format. Furthermore, new 'partial correlation analyses' should be performed, adjusting for age, sex, BMI, diabetes, CAD, hypertension and eGFR.

Table 1: should spell out the sample size of HCM, no DD, and DD groups.

Author Response

Response to Reviewer 1

Introduction:      should spell out that the ADMA and SDMA have diverse effects on NO      synthesis.

We outlined the specific effects of ADMA and SDMA in the Introduction section in Line 45-46

Introduction: the biological      role of homoarginine, e.g. arginase inhibition, is far from established      (e.g. Tommasi S et al, Sci Rep 2018;8(1): 3697). This should be discussed.

We discussed this point in line 49 – 51 in the Introduction section

Methods, CMR: this paragraph      should spell out which specific parameters were measured for the purpose      of the analysis.

Specific MRI parameters were selected for the analysis. Ejection fraction, stroke volume, left ventricular mass, septal wall, fibrosis detection (yes or no), fibrosis in 5 SD and myocardial contraction fraction (MCF) were determined. The first four parameters were already determined in echocardiography, so the focus was on fibrosis and MCF.

This is now mentioned in Methods 2.5.

Methods: the authors should clearly describe the study hypothesis and list the primary and secondary      end-points of cardiac structure and function.

The primary and secondary endpoints of our study are outlined now at the end of the Introduction section (line 54-57) – if you wish to integrate this in the Methods section instead, we can do so.

A key issue that limits data interpretation is the lack of adjustment for eGFR. ADMA and, particularly, SDMA undergo renal elimination. I note that eGFR is higher in no DD than DD patients (Table 1). All regression and ANOVA analyses should also be adjusted for eGFR. 

We thank the reviewer for this comment and calculated for all regression analyses model 4, that is additionally adjusted for the eGFR. For ADMA all associations remained significant (logistic regression comparing HCM patient with moderate to severe DD and HCM patients with normal diastolic function, Table 3; linear regression of arginine derivatives and echocardiographic parameters of diastolic function, Suppl. Table 4). The association of SDMA and AF in HCM patients missed significance after further adjustment for eGFR (logistic regression analyses, table 4). eGFR-adjusted ANCOVA revealed a differentiation of patients with a normal diastolic function and patients with a moderate to severe DD by ADMA (P=0.026). No differences were found for SDMA and hArg. Adjustments in the manuscript can be found in paragraph 2.6, 3.1, 3.2, table 3 and supplement table 4

Present all correlation analyses in Table format. Furthermore, new 'partial correlation analyses' should be performed, adjusting for age, sex, BMI, diabetes, CAD, hypertension and eGFR.

We added a table with all Spearman correlation analyses (Suppl. Table 3) and additionally performed Pearson and partial correlation analyses where no associations were found.

Table 1: should spell out the      sample size of HCM, no DD, and DD groups.

We added the sample sizes in Table 1 (HCM: n=215; HCM no DD: n=61, and HCM DD: n=143).

Reviewer 2 Report

Paper is interesting but some revision are needed before it can be accepted:

- In methods section authors firstly described theyr studies as a retrospective one and then as a prospective one. Please clarify and, if it is a retrospective one, I would like to know if patients where recalled for the sign on the informed consent or wether. If it is a prospective one did from when to when data were recorder? Another possibility could also be, i.e that patients were retrospectively individuated in your previous cohort and they were recalled for the assessment of NO correlates while retrospective echo and RMN data were used. Please clarify these fundamental point.

- Only 57 patients were genetically confirmed in this cohort so maybe other causes of HCM where included. A different analysis on this 57 patients to confirm the data of the whole cohort need to be done and to be described as supplementary material. Please enlist this point in the limitation section.

- Model 3 of linear regression model presents arterial hypertension as a covariates. Are continuous BP data available? if yes please use it in the model instead of the diagnosys of hypertensione. If not enlist it as a limitation in the relative section.

- The main objective of the study is declared to be evaluation of the correlation between NO product and DD while also part on AF and left ventricular hypertrophy are present in the results. Please add this point to the objective of your study.

- Please show also the correlation with the LGE at the RMN.

- Some studies on endothelial function and DD in HCM patients have been missed (as an example

Peripheral microvascular function is altered in young individuals at risk for hypertrophic cardiomyopathy and correlates with myocardial diastolic function.     Fernlund E et al. Am J Physiol Heart Circ Physiol.         (2015)  

Vascular Endothelial Growth Factor Is Associated with the Morphologic and Functional Parameters in Patients with Hypertrophic Cardiomyopathy.     Pudil R et al. Biomed Res Int.         (2015)  ). They need to be cited and commented in the discussion section.

- Furthermore, NO metabolism is strongly related to inflammation. Are any inflammatory biomarkers available?

- On the same way endothelial function has been found to be impaired in inflammatory condition such the one described in the following paper (

J Antimicrob Chemother. 2018 Aug 1;73(8):2162-2170. doi: 10.1093/jac/dky178.

Evaluation of adhesion molecules and immune parameters in HIV-infected patients treated with an atazanavir/ritonavir- compared with a lopinavir/ritonavir-based regimen.

Squillace N1, Trabattoni D2, Muscatello A1, Sabbatini F1, Maloberti A3, Giannattasio C3, Masetti M2,4, Fenizia C2, Soria A1, Clerici M4, Gori A1, Bandera A1.

). Please cite and comment the possibile influence of inflammatory factor in HCM and their possible role in DD in those patients.

Author Response

Response to Reviewer 2

In methods section authors firstly      described theyr studies as a retrospective one and then as a prospective      one. Please clarify and, if it is a retrospective one, I would like to      know if patients where recalled for the sign on the informed consent or      wether. If it is a prospective one did from when to when data were      recorder? Another possibility could also be, i.e that patients were      retrospectively individuated in your previous cohort and they were      recalled for the assessment of NO correlates while retrospective echo and      RMN data were used. Please clarify these fundamental point.

The present study is retrospective.

Patients came at regular intervals for follow-up visits. At the first appointment they signed the study protocol which allowed us to use their blood samples and imaging data also from follow up visits for specific research approaches. We tried to use the first data collection for all patients. However, we made sure that blood samples and imaging were performed within a 6 month frame.

Only 57 patients were genetically confirmed      in this cohort so maybe other causes of HCM where included. A different      analysis on this 57 patients to confirm the data of the whole cohort need      to be done and to be described as supplementary material. Please enlist      this point in the limitation section.

We did further analyses and included the following into the manuscript:

4.4. Study limitations: …” only 56 out of 215 were genetically confirmed HCM patients”…

  3.1. Arginine derivatives and diastolic dysfunction:
  “In the subgroup of the genetically confirmed HCM-patients ADMA concentration         also increased with increasing severity of DD (Suppl. Table 1).”
  “In unadjusted regression analyses ADMA concentration showed an association when        comparing genetically confirmed HCM patients with a moderate to severe DD with patients having a normal diastolic function (Suppl. Table 2).”
  “Correlation and linear regression analyses were not reproducible in the small subgroup of genetically confirmed HCM patients (data not shown).”

            3.2. Arginine derivatives and atrial fibrillation:
            “In the subgroup of genetically confirmed HCM patients SDMA was also associated with AF in unadjusted logistic regression analyses (Suppl. Table 5).”

Model 3 of linear regression model presents arterial hypertension as a covariates. Are continuous BP data available? if yes please use it in the model instead of the diagnosys of hypertensione. If not enlist it as a limitation in the relative section.

Data on continuous BP are not available.

The main objective of the study is declared to be evaluation of the correlation between NO product and DD while also part on AF and left ventricular hypertrophy are present in the results. Please add this point to the objective of your study.

The main objective of the study is to evaluate the correlation between NO products and DD. However, as a second objective, we considered the correlation between Arg derivatives and fibrosis determined as LGE in CMR, the presence of AF, and left ventricular hypertrophy as outlined in line 55-57. 

Please show also the correlation with the LGE at the RMN.

We additionally performed correlation analyses of arginine derivatives with LGE and did not see any associations (Suppl. Table 1).

Some studies on endothelial function and DD in HCM patients have been missed (as an example

Peripheral microvascular function is altered in young individuals at risk for hypertrophic cardiomyopathy and correlates with myocardial diastolic function.     Fernlund E et al. Am J Physiol Heart Circ Physiol.         (2015)  

Vascular Endothelial Growth Factor Is Associated with the Morphologic and Functional Parameters in Patients with Hypertrophic Cardiomyopathy.     Pudil R et al. Biomed Res Int.         (2015)  ). They need to be cited and commented in the discussion section.

These studies are included (Line 40/41 and Line 214-216).

Furthermore, NO metabolism is strongly related to inflammation. Are any inflammatory biomarkers available?

In our study inflammatory biomarkers were not addressed and we gave a statement in the limitation section (line 256-257). We therefore decided not to cite any related studies.  But we agree with the reviewer, that this is an important factor, which should be analysed in more detail in future studies.

Round 2

Reviewer 1 Report

- Results: Please delete the following sentence "In the subgroup of genetically confirmed HCM patients SDMA was also associated with AF in unadjusted logistic regression analyses." Unadjusted analysis is not relevant.

- Conclusions: "Higher SDMA and lower hArg were associated with AF in our cohort". SDMA was not independently associated with AF after adjusting for eGFR (Table 4). Please revise.

Author Response

- Results: Please delete the following sentence "In the subgroup of genetically confirmed HCM patients SDMA was also associated with AF in unadjusted logistic regression analyses." Unadjusted analysis is not relevant.

According to the reviewer`s suggestion we deleted the sentence and wrote instead “No associations of arginine derivatives were found in adjusted logistic regression analyses in the subgroup of genetically confirmed HCM patients (Suppl. Table 5).“ (section 3.2)

- Conclusions: "Higher SDMA and lower hArg were associated with AF in our cohort". SDMA was not independently associated with AF after adjusting for eGFR (Table 4). Please revise.

We revised this point in section 4.2 (discussion) and section 5. (conclusion).

4.2: “In our HCM cohort decreased hArg was independently associated with the presence of AF within the last five years prior to inclusion in this study.”

5. “Lower hArg was associated with AF in our cohort […].”

Reviewer 2 Report

Paper can now been accepted

Author Response

no further requests